# Utility of a patient similarity-based digital tool for risk communication to patients with type 2 diabetes mellitus: perspectives from primary care physicians in ambulatory care

**Ruiheng Ong**[1]*, **Chirk Jenn Ng**[1,2], **Kalaipriya Gunasekaran**[1], **Hang Liu**[3], **Wynne Hsu**[3], **Mong Li Lee**[3], **Ngiap Chuan Tan**[1,2]

**1** SingHealth Polyclinics, SingHealth, Singapore, Singapore, **2** Family Medicine Academic Clinical Programme, SingHealth-Duke NUS Academic Medical Centre, Singapore, Singapore, **3** Institute of Data Science, National University of Singapore, Singapore, Singapore

* ong.ruiheng@singhealth.com.sg

## Abstract

### Background

Inaccurate risk perceptions of diabetes complications are responsible for the inertia among patients to engage in protective health behaviours. One potential approach to changing risk perceptions is to use social comparison of their diabetes to other people of similar clinicodemographic profiles.

### Objectives

This study examined the perspectives of primary care physicians (PCPs) in ambulatory care on the utility of a patient similarity-based digital tool for risk communication to patients with type 2 diabetes mellitus (T2DM).

### Methods

A qualitative study design using direct observation and in-depth interviews was conducted on 11 PCP participants. Participants had at least 6 months of clinical experience in ambulatory primary care. Participants went through three hypothetical case scenarios using the digital tool under direct observation and shared their perspectives on its utility during an in-depth interview. Data were coded and analysed using thematic analysis.

### Results

PCPs perceived the digital tool to be useful in educating patients with newly diagnosed or uncontrolled T2DM and to motivate them to achieve better glycated haemoglobin (HBA1c) levels. Patients who do not practise social comparison would refrain from HBA1c comparison and prefer to know the absolute state of their diabetes. PCPs were also concerned about patients' potential for false reassurance or negative reactions instead of correctly understanding the risk message intended for them.

**Data availability statement:** All relevant data are within the manuscript and its Supporting Information files.

**Funding:** WH, MLL and NCT received funding from the AI in Health Grand Challenge (Award Number: AISG-GC-2019-001) by AI Singapore (AISG) under the National Research Foundation Singapore. The URL of the funder website is aisingapore.org/jarvisdhl-transforming-chronic-care-for-diabetes-hypertension-and-hyperlipidemia-dhl-with-ai. The sponsors or funders did not play any role in the study design, data collection and analysis, decision to publish, or preparation of the manuscript.

**Competing interests:** I have read the journal's policy and the authors of this manuscript have the following competing interests: WH, MLL and NCT are Principal Investigator and Co-Principal Investigators respectively of the project "JARVIS-DHL: Transforming Chronic Care for Diabetes, Hypertension and HyperLipidemia (DHL) with AI" which is funded by the AI in Health Grand Challenge (Award Number: AISG-GC-2019-001) by AI Singapore (AISG) under the National Research Foundation Singapore. This does not alter our adherence to PLOS ONE policies on sharing data and materials.

## Conclusion

The patient similarity-based digital tool requires further work to support PCPs in risk communication to patients with T2DM. Usage should be targeted at patient subgroups with newly diagnosed or uncontrolled T2DM and who practise social comparison. Strategies to maximise benefit include identifying patients who practise social comparison and training PCPs to be adept at framing and communicating risk information in a person-centric manner to mitigate the possibility of false reassurance or negative reactions from their patients.

## Introduction

Accurate risk perception is pivotal in driving protective health behaviours [1–3]. Among people with type 2 diabetes mellitus (T2DM), inaccurate perceptions of the risks of diabetes complications are responsible for the inertia to engage in protective health behaviours [4–8]. Risk communication interventions for healthcare professionals to convey quantitative risk information of diabetes complications to patients, especially for cardiovascular disease, have been studied with mixed results [9–15]. For example, French et al. demonstrated that presenting risk information both numerically and graphically improved the accuracy of patients' risk perceptions but did not lead to behavioural change [10]. Charlson et al. demonstrated no difference in major adverse cardiovascular events when benefits of cardiovascular risk factor modification were framed to patients using present biological age reduction, compared to using future cardiovascular risk reduction [11]. Grover et al. demonstrated that patients were more likely to achieve their lipid targets when their individual cardiovascular risk was presented to them as a 'cardiovascular age' (comparing to people of the same age and sex without modifiable risk factors); however, this was only applicable to patients with no pre-existing cardiovascular disease [12]. These highlight a growing need to bridge the knowledge gap on how healthcare professionals can convey risk information in meaningful ways to patients to effectively motivate protective behaviours to reduce the risk of diabetes complications [9].

One potential approach to changing an individual's risk perception is to use social comparison, whereby a person's own health is compared to other people of a similar demographic and clinical profile [16–18]. In patients with chronic diseases, social comparison of their health has been found to mediate the perception of their health and influence health-related decisions [19–22]. Recent research has reported superiority of patient similarity-based models over general population-based models in predicting health outcomes [23–25], underpinned by the concept that "similar patients with similar features have similar outcomes" [26]. Martinez et al alluded that patients with T2DM valued being compared to patients of similar demographics (age group, gender, ethnicity, and body type) and on similar medications, and hence were more likely to increase their motivation for self-care [27].

Recent advances in Artificial Intelligence (AI) technology, particularly in using machine learning on big datasets, have provided opportunities to predict health outcomes of T2DM patients more precisely by harnessing real-world data extracted from the electronic health records (EHR) [28–30]. In Singapore, a team of primary care physicians (PCPs) and data scientists developed an AI-driven patient similarity model named D3K (domain knowledge and data-driven insights) and demonstrated its reliability in predicting complications among patients with diabetes, hypertension and/or hyperlipidemia, when compared to existing clinical risk prediction models [31,32]. A digital tool, named PERDICT.AI (PERsonalised DIabetes Counselling Tool using Artificial Intelligence), was designed based on this model, and its target users are PCPs in ambulatory care. Using Social Comparison Theory [16–18]

and big data analytics, the patient similarity model generates risk information reflective of algorithmically-similar T2DM patients from the same cluster of primary care clinics (polyclinics) at the time of data extraction. This is intended to support PCPs in risk communication during their consultations with T2DM patients.

In addition to being competent in using the digital tool during consultation with patients, PCPs will need to know how to tactfully frame and communicate risk information in a skilful manner to successfully change their patients' risk perception and motivate them to take up protective health behaviours [33,34]. Hence, PCPs' receptivity towards PERDICT.AI's patient similarity-based approach for risk communication to patients is crucial for further refinement of the digital tool and its adoption into clinical practice. This study therefore aimed to examine the perspectives of PCPs in ambulatory care on the utility of the patient similarity-based digital tool for risk communication to patients with T2DM. The results will identify their perceived barriers and challenges encountered when using the tool. The feedback from PCPs will help inform further revisions to the tool and address barriers to adoption in an ambulatory primary care setting.

## Methods

### Study design and setting

A qualitative study design was used to explore the utility of the patient similarity-based digital tool from the PCP's perspective for risk communication to T2DM patients. This consisted of direct observation and in-depth interviews, conducted between September 2022 and January 2023 in three public primary care clinics (polyclinics) within a healthcare cluster in the Eastern region of Singapore. In total, PCPs in this healthcare cluster provide ambulatory primary care for more than 200,000 patients with T2DM per year. This includes the diagnosis of T2DM, monitoring of glycaemic control, and the initiation and adjustment of anti-diabetic agents and insulin. The study was part of a larger study to evaluate usability and utility of the digital tool.

Ethics approval was obtained from the SingHealth Centralised Institution Review Board (CIRB 2021/2661). Informed written consent was obtained from all participants prior to participation in the study. The study followed the consolidated criteria for reporting qualitative studies (COREQ) (S1 Appendix) [35].

### Study team and reflexivity

The core study team comprised three male PCPs – RO (MMed), CJN (PhD) and NCT (MMed) – who are qualified and actively-practicing Family Physicians, and one female Research Associate, KG (MD), who formerly practised as a physician in Community Health and Preventive Medicine. RO, KG, CJN and NCT are trained in qualitative research, while CJN and NCT are experienced primary care researchers and hold professorial appointments in Family Medicine. Additional support to provide iterative revisions of the PERDICT.AI tool was provided by three female computer scientists – HL (MSc), WH (PhD) and MLL (PhD) – who were instrumental in designing the tool together with NCT. RO, CJN and NCT performed alpha testing of PERDICT.AI prior to the study to familiarise themselves with the technical aspects of using the tool.

As a practising PCP in a clinic with a high volume of patient visits, RO was cognisant of barriers that PCPs practising in similar environments would face towards adopting new technology. RO reflected on his dual role as a researcher and clinician, and had ongoing discussions on these with CJN and NCT. RO made concerted efforts to switch his attention away from his clinician role and immersed himself into the researcher role to maintain distance

from his personal views during data collection and analysis. RO remained receptive to understanding positive and negative perspectives given by the PCP participants. PCP participants, especially those of lower seniority to RO, were encouraged during the interviews to share their honest responses and highlight issues faced during usage of PERDICT.AI that would inform the study team on areas to improve the tool. Open-ended questions were used, and leading questions were avoided. Active listening and verbal cues were used to encourage participants to articulate their views. Where needed, paraphrasing and clarification was done to ensure accurate understanding of participants' responses. CJN was present when RO conducted the first interview, and a debriefing was done post-session.

## Participants and sampling

Participants were PCPs aged 21 years and above and had least 6 months of clinical experience in ambulatory primary care. PCPs were recruited by purposive sampling with maximum variation to capture a diverse range of seniorities, with participants representing early-career (Medical Officers), mid-career (Family Physicians) and late-career (Consultants) stages. Recruitment was done between 6 September 2022 to 6 January 2023. Sixteen potential PCP participants were sent email invitations with a user guide on PERDICT.AI and subsequently approached at the study sites for face-to-face recruitment. The sample size was guided by Hennick and Kaiser who found that saturation is often achieved within 9-17 interviews [36]. RO introduced himself to each PCP participant as the Principal Investigator of the study and explained the PERDICT.AI tool and purpose of the study. Assurance was given about data confidentiality. Eleven PCPs consented to participate, and five declined to participate citing unavailability of time. Two of the eleven PCP participants had no prior relationship with RO. Two PCPs have taken part in clinical workgroup activities with RO in a non-academic setting. The remaining seven PCPs knew RO only at an acquaintance level due to the relatively smaller number of Family Physicians within the institution. Most of the PCPs knew CJN and NCT as academic appointment holders in the institution's research department. Training on using PERDICT.AI was not conducted, as the intent was for participants to use PERDICT.AI for the first time as a naïve user. The study was conducted over two iterative cycles. Five and six PCPs took part in the first and second cycles, respectively. Usability issues of PERDICT.AI in the first cycle were rectified, and a revised version of PERDICT.AI was used in the second cycle. According to Nielsen and Virzi, at least 5 participants are required to identify most of the usability issues during user testing [37,38].

## Research instruments

The PCP participants completed a standardised questionnaire to record their baseline demographic details. An observation log was used by the investigators to document observations about participants' PERDICT.AI usage and any comments participants raised during the session. An in-depth (semi-structured) interview topic guide (S2 Appendix) was developed based on Smart's multidimensional model of clinical utility, which takes into consideration the care provider's view on appropriateness and acceptability of the clinical intervention of interest [39].

## PERDICT.AI

The PERDICT.AI digital tool (S3 Appendix) compares a patient's glycated haemoglobin (HBA1c) levels to other T2DM patients from the same cluster of primary care clinics (polyclinics) and displays the aggregate prevalence of diabetes complications among T2DM patients of similar demographic and clinical profiles. Upon entering a given patient's

non-identifiable details into PERDICT.AI, the patient similarity algorithm searches through a de-identified T2DM database to identify 30 patients with the highest similarity scores, referred to as the "similar patient subgroup." The aggregate 5-year incidence of a given diabetes complication among the group is computed from the database (S4 Appendix). Going a step further, PCPs can illustrate negative and positive case examples to their patients (S5 Appendix) [31,32].

For example, using a male patient who is aged xx years with a duration of T2DM of xx years as a used case for illustration, the participant was able to view the clinical profiles of a group of 30 other similar patients. The group's aggregate 5-year incidences of macrovascular, kidney, eye and foot complications are numerically and graphically represented (S4 Appendix). Case examples illustrate how patients whose HBA1c (glycated haemoglobin; haemoglobin A1c) decreases over time are less likely to develop complications, while patients whose HBA1c remains high or increases over time are more likely to develop complications over the next 5 years (S5 Appendix). The message to patients is clear: "As alluded by similar patients who are similar to you, 3 in 10 would develop blood vessel disease, 6 in 10 kidney disease, 2 in 10 eye disease and 1 in 10 foot problems."

## Data collection

Data collection was conducted by RO face-to-face with the participant in a quiet room within the study site clinic for the convenience of the participants. No non-participants were present. English language was used as the form of communication as all participants were proficient in spoken and written English. PCP participants used PERDICT.AI on a laptop and went through three hypothetical patient case scenarios (S6 Appendix) under direct observation, keying in relevant details into PERDICT.AI and interpreting the risk information it generated. An observation log was used to document observations about participants' PERDICT.AI usage and any comments participants raised during the session. Each participant took between 17 and 48 minutes to complete the case scenarios. After completion of the case scenarios, in-depth interviews with retrospective probing were conducted by RO to examine their perspectives on the utility of PERDICT.AI and to identify the challenges encountered. Each interview took 30 to 60 minutes and was audio-recorded. Field notes were taken during the session and at the end of the interview. There were no repeat interviews. All eleven participants completed the study. A flow diagram of the data collection is illustrated in Fig 1.

## Data analysis

All audio recordings were transcribed verbatim and checked for completeness and accuracy. RO and CJN familiarised themselves with the transcripts from the first two in-depth interviews and independently coded the transcripts. A list of open codes was assigned to the transcripts based on the study objective; the open codes were combined to form a coding

Participant's baseline details obtained
↓
Direct observation of participant's PERDICT.AI usage with 3 hypothetical patient case scenarios
↓
In-depth interview with participant

**Fig 1. Flow diagram of data collection.**

framework. RO and CJN met to reached a consensus on the coding framework. RO coded the remaining transcripts and new codes were added iteratively after discussion with the study team. The data was managed using NVivo Windows Release 1.5.1. Emergent themes were identified from the data (Table 1). Thematic saturation was reached after 9 interviews. No new themes emerged from the last 2 interviews.

## Results

### Participant characteristics

A total of eleven PCPs between age 28-57 years participated in the study. Characteristics of participants are detailed in Table 2.

### Principal findings

Three major themes on utility emerged from the data: (1) education and motivation for subgroups of patients with T2DM; (2) patients who do not practise social comparison; and (3) potential for false reassurance or negative reactions from patients.

**Theme 1: Education and motivation for subgroups of patients with T2DM.** PCPs perceived the digital tool to be useful in educating patients about the state of their diabetes and forewarning them about the consequences of uncontrolled diabetes using real case examples. For patients with high HBA1c, comparing HBA1c with other patients can motivate them to achieve better HBA1c levels than what they currently believe to be acceptable. Patient subgroups likely to benefit are those with newly diagnosed or uncontrolled T2DM.

*"I think that (the tool can be used) for newly diagnosed diabetes patients for patient education to show them real examples. I actually want to use a negative example, a generic negative example and positive example to show the patient at the start when they're first diagnosed with diabetes, how this could go, and the importance of control... the other option is I might use it for patients who have poorly controlled diabetes and think that nothing will happen to them... and then I can show them the examples."*

*-Participant 2, 29-year-old Family Physician with 1 years' experience in primary care*

Table 1.  Emergent themes and their respective codes.

| Theme | Codes |
|---|---|
| Education and motivation for subgroups of patients with T2DM | • Corrects wrong beliefs |
| | • Educated patient |
| | • Motivates improvement |
| | • Newly diagnosed DM |
| | • Objective examples |
| | • Patients who socially compare |
| | • Uncontrolled DM |
| | • Wake-up call |
| Patients who do not practise social comparison | • Information needs to be individualised |
| | • Patients who do not socially compare |
| Potential for false reassurance or negative reactions from patients | • Discouragement |
| | • False reassurance |
| | • Reinforces wrong beliefs |
| | • Worry excessively |

*"So, it (HBA1c comparison) can keep them (patients) motivated if they are doing well or if they are lagging behind and then it can serve as a motivation to get better… I guess that depends on patient profile… I think for the start, the earlier stages, it's quite useful to compare, so that we can catch them early (before their condition worsens)."*

-Participant 6, 29-year-old Family Physician with 5 years' experience in primary care

*"I think this one (HBA1c comparison) is quite useful… the patient knows that they are higher than half of the patients of their own age… they don't feel that it's actually normal because some patients come in and say that, 'Oh, my friends say that (a HBA1c of) more than ten (percent) is very bad, it is okay'. So, at least this one tells them the truth."*

-Participant 3, 32-year-old Family Physician with 3 years' experience in primary care

*"It (PERDICT.AI) gives you some idea of where you stand in terms of diabetic control… with regards to other people. It kind of gamifies it… I do think that there are some people out there who actually would want to feel like they're much healthier or they do much better than the average population."*

-Participant 10, 28-year-old Medical Officer with 6 months' experience in primary care

*"In fact a lot of patients know what are the complications and their impact… they are often not convinced on the need to change… maybe something like this might push or nudge (them) a bit more… because sometimes they might think we are just setting some unfair targets right?"*

-Participant 11, 36-year-old Consultant with 8 years' experience in primary care

**Table 2. Characteristics of participants (N = 11).**

| Participant characteristic | n (%) |
|---|---|
| **Sex** | |
| Male | 6 (54.5) |
| Female | 5 (45.5) |
| **Age (years)** | |
| < 30 | 4 (36.4) |
| 31-35 | 2 (18.2) |
| 36-40 | 2 (18.2) |
| 41 or more | 3 (27.3) |
| **Seniority** | |
| Medical Officer | 2 (18.2) |
| Family Physician | 4 (36.4) |
| Consultant | 5 (45.5) |
| **Clinical experience in ambulatory primary care (years)** | |
| < 1 | 2 (18.2) |
| 1-5 | 3 (27.3) |
| 6-10 | 3 (27.3) |
| 11 or more | 3 (27.3) |
| **Participant characteristic** | **Mean (σ)** |
| Age (years) | 36.5 (9.2) |
| Clinical experience in ambulatory primary care (years) | 8.0 (7.4) |

**Theme 2: Patients who do not practise social comparison.**  PCPs expressed concern that some patients do not practise social comparison; these patients refrain from comparing their HBA1c with other patients and would prefer to know the absolute state of their diabetes. Some PCPs perceived the need for HBA1c targets to be individualised to the patient, instead of going by the basis of a cohort ranking.

*"I think it's the absolute (HBA1c) target and the risk (of complications that matters). So, I think that is how we actually… that's how I buy in the patients."*

*-Participant 1, 57-year-old Consultant with 25 years' experience in primary care*

*"I'm not sure whether it's really important to differentiate (patients' HBA1c levels) between their age group and all patients… in terms of HBA1c target per say, we may also need to individualise it a little bit especially for those with CKD (chronic kidney disease) and those who have like very frail recurrent hypoglycaemia… I probably would want to have (the patient's) HBA1c target here to say that specifically for you based on your risk factors, your HBA1c target is this."*

*-Participant 2, 29-year-old Family Physician with 1 years' experience in primary care*

*"I always try to personalise things for my patients… for me it doesn't matter what the rate of complications (among the similar patient cohort) is… (or) how much higher the HBA1c of my patient is compared to the rest of the population because it's not going to change the way I titrate his or her medications. It's not going to change the way I counsel the patient about diet, lifestyle or any other modifications."*

*-Participant 5, 36-year-old Family Physician with 8 years' experience in primary care*

*"I think (we should be) more specific when we counsel patients… something more targeted to them would be better to counsel with. We always need to tell them (patients) what the actual thing (HBA1c) is: 'The HBA1c should be less than seven… but in your case since there is a lot of hypoglycaemia then maybe it's more dangerous so, we (need) to balance this… we will set your target as 7.5 or 8.' But they must know that eight is already a stretch from safety because of a counterbalance of adverse events."*

*-Participant 7, 47-year-old Consultant with 14 years' experience in primary care*

*"I think the absolute (HBA1c) is more important than the cohort(-based comparison)… there are a lot of non-social people… and that's why peer nudging doesn't work for a lot of people… 'My life is my life… I don't care that others are better or worse off than me.' It's not a competition… I think that there are actually many people out there who will not be influenced (by peer nudging)… they do what they do and they feel what they feel."*

*-Participant 9, 47-year-old Consultant with 17 years' experience in primary care*

**Theme 3: Potential for false reassurance or negative reactions from patients.**  PCPs were concerned that patients could react to the risk information differently from what the PCP intends, instead of correctly understanding the risk message intended for them, thereby hindering PCPs from using the tool. Patients could develop false reassurance by perceiving an artificially more favourable state of diabetes for themselves – for example, by focusing on the subgroup with poorer glycaemic control instead of the subgroup with better glycaemic control, or by focusing on the majority without diabetes complications.

*"To me, I think that in terms of explaining to the patient's their treatment goals, it should be individualised. But then conveying it to the patient, sometimes… if they see that their HBA1c is worse than (other) people, it may be easier for them to understand… but then, I also don't want to give them this false sense of reassurance that oh, just because it's better (than other people) doesn't mean you have hit the target (HBA1c) yet."*

*-Participant 2, 29-year-old Family Physician with 1 years' experience in primary care*

*"Many patients would feel that 'The risk of foot complications is quite low so I don't really need to bother about that,' but actually that can lead to amputations which are also quite serious complications."*

*-Participant 8, 32-year-old Consultant with 6 years' experience in primary care*

*"A person with diabetes on diet control is still at high (cardiovascular) risk as (compared to) a person with no diabetes… and to say that 'Oh, you're in the majority (for having relatively better HBA1c levels)… that means you're good, right?' No, I won't even use this at all… the messaging is not clear… If you're trying to use (the pie chart representation of diabetes complications among) peers to pressure me then I say, 'But I'm not them (smaller red sector with complications), maybe I'm the pink one (larger pink sector without complications). In other words, most people don't have (complications and) that means I'm in the pink. So, I don't care, right… now I'm in the majority, so won't happen to me.' But if… instinctively, I say that your risk is higher… and the red (sector) represents that risk, then I see (that) 'Oh (I am at) risk, risk to me'. To the patient… it's very visual. It's what you take away from a picture… all they see is that 'Okay, I see a bigger red sector' (or) 'I see a smaller red sector'. And… if I take this to be my risk of getting complications, then a big red (sector) stands out more to me."*

*-Participant 9, 47-year-old Consultant with 17 years' experience in primary care*

*"I wonder how this data would change if let's say, it's (the) elderly and poorly controlled DM (diabetes mellitus) (that a patient's HBA1c is being compared against). If all of them have poor controlled DM (diabetes mellitus), then… I also don't want the data to work against me. Like let's say, I still want a lower target but it (HBA1c comparison) shows that actually he is okay."*

*-Participant 11, 36-year-old Consultant with 8 years' experience in primary care*

Conversely, patients could be discouraged or unduly distressed when informed about the unfavourable state of their diabetes – for example, if due to circumstances out of their control – thereby leaving a negative experience in their healthcare journey.

*"Certain patients will tend to prefer not to hear bad news. If you tell them their sugar control is bad, they will say they are worried or angry or upset. A patient may be quite affected by their control. Maybe it's out of their control (for example due to) recent infection… it might just add to their negative experience of healthcare and diabetes control."*

*-Participant 10, 28-year-old Medical Officer with 6 months' experience in primary care*

## Discussion

The emergent themes reflect key components of Smart's model of utility, which is a "multi-dimensional judgement about the usefulness, benefits, and drawbacks of an intervention" – namely appropriateness and acceptability [39]. Appropriateness – comprising efficacy and

relevance to the target population – is reflected by the themes "Education and motivation for subgroups of patients with T2DM" and "Patients who do not practise social comparison." Acceptability – comprising stakeholder perspectives such as those of care providers and patients – is reflected by the theme "Potential for false reassurance or negative reactions from patients." The model's remaining key components – practicability and accessibility – are beyond the scope of this study. Apart from widely reported outcome-based measures of utility such as effectiveness and cost-effectiveness, the above components are equally crucial for new clinical interventions to be purposefully adopted into clinical practice.

From the study findings, the patient similarity-based digital tool requires further work to achieve its potential to support PCPs in strengthening patients' awareness of their diabetes control and complication risks. To maximise benefit within the limitations of short consultation times, usage of the tool will need to be targeted at subgroups of T2DM patients with newly diagnosed or uncontrolled T2DM and who practise social comparison. Questionnaires measuring social comparison orientation, such as INCOM (Iowa-Netherlands Comparison Orientation Measure) [40], can be locally adapted and validated for T2DM patients. To our best knowledge, this has not yet been studied in the context of clinical interventions.

PCPs were primarily concerned about the potential for patients to develop false reassurance or negative reactions from selective processing of the risk information. To address this, training must be provided for PCPs to anticipate such scenarios and to be adept at emphasising relevant details so that patients have a better chance of recognising the unfavourable state of their diabetes to avoid being falsely reassured – all this while maintaining accuracy and objectivity of the information provided. It is important to emphasise during PCP training that the prevalence rates of diabetes complications among the "similar patient subgroup" is a cross-sectional snapshot of this subgroup which the patient is similar in terms of clinical profile and demographics; it should not be interpreted as an individual's predicted risk of developing these complications. Validated risk score calculators should still be used for such predictions, such as the UK Prospective Diabetes Study (UKPDS) [41] in the UK and Framingham Adult Treatment Panel (ATP) III [42] in Singapore for cardiovascular disease risk calculation. Finally, improving PCPs' user experience of the digital tool will also be crucial to their receptiveness towards adapting the tool into their clinical practice to facilitate risk communication.

The digital tool's effectiveness in changing patient's risk perceptions will depend on the patient's tendency to practise social comparison and the PCP's ability to anticipate false reassurance or negative reactions and to be adept at framing and communicating risk information in a person-centric manner. The study findings provide some insight into how PCPs may be empowered to leverage on social comparison to position patients to compare their diabetes with other patients, advance their risk awareness and promote health protective behaviour to achieve better diabetes control [43]. This underscores the importance of facilitating doctor-patient dialogues using the risk information in PERDICT.AI instead of "climbing probability trees" [44,45].

This study has some limitations. The study findings were not given to participants for feedback. However, these were cross-checked with the observations about participants' PERDICT. AI usage and any comments participants raised during the session. As there are no patient participants, the findings are based solely on doctors' perspectives. The authors will embark on a follow-up study involving patient participants to triangulate the findings. Generalisability is restricted due to the study design.

There are limitations to the digital tool's retrospective approach to identifying algorithmically similar patients through a database. Some of the variables of interest exist as unstructured data or data on these variables could be incomplete or insufficient [46] – for example, medication adverse events and hypoglycaemia episodes, which are crucial for clinical decision-making and require

doctors' clinical acumen to elicit before sizeable data can be captured prospectively and integrated into the similarity algorithm [47]. Consideration can be made to use trajectory analysis (i.e., multiple timepoints) of HBA1c instead of the current cross-sectional (i.e., single timepoint) approach.

## Conclusion

The patient similarity-based digital tool, PERDICT.AI, requires further work to achieve its potential to support PCPs in ambulatory care for risk communication to patients with T2DM. Usage of the tool should be targeted at patient subgroups with newly diagnosed or uncontrolled T2DM and who practise social comparison. Strategies needed to maximise benefit from the digital tool include identifying patients who practise social comparison and training PCPs to be adept at framing and communicating risk information in a person-centric manner to mitigate the possibility of false reassurance or negative reactions from their patients. Further research can be done to evaluate the feasibility and acceptability of the clinical intervention among this subgroup of T2DM patients.

## Supporting information

**S1 Appendix. COREQ checklist.**
(PDF)

**S2 Appendix. Topic guide.** This was used during in-depth interviews to explore participants' views on the appropriateness and acceptability of the PERDICT.AI digital tool and to identify the challenges encountered.
(PDF)

**S3 Appendix. PERDICT.AI digital tool.** This is a detailed description of modules in the PERDICT.AI digital tool.
(PDF)

**S4 Appendix. Example of an aggregate 5-year incidence of diabetes complications among the "similar patient subgroup".**
(TIF)

**S5 Appendix. Example of a pair of negative and positive case examples from the "similar patient subgroup".**
(TIF)

**S6 Appendix. Hypothetical patient case scenarios for use in the PERDICT.AI digital tool** .
(PDF)

## Acknowledgements

The authors would like to acknowledge Wei Ying Tan from the Institute of Data Science, National University of Singapore for her work in developing the patient similarity algorithm used in PERDICT.AI, as well as Patricia T Kin, Yang Thong Tan, Usha Sankari, Paulpandi Muthulakshmi, Jullina Binte Buang and Ai Choo Seah from the Department of Research, SingHealth Polyclinics for their support in making this work possible.

## Author contributions

**Conceptualization:** Ruiheng Ong, Chirk Jenn Ng, Hang Liu, Wynne Hsu, Mong Li Lee, Ngiap Chuan Tan.

**Formal analysis:** Ruiheng Ong, Chirk Jenn Ng, Kalaipriya Gunasekaran, Ngiap Chuan Tan.

**Investigation:** Ruiheng Ong, Chirk Jenn Ng, Ngiap Chuan Tan.

**Methodology:** Ruiheng Ong, Chirk Jenn Ng, Ngiap Chuan Tan.

**Software:** Hang Liu, Wynne Hsu, Mong Li Lee.

**Supervision:** Chirk Jenn Ng, Ngiap Chuan Tan.

**Writing – original draft:** Ruiheng Ong.

**Writing – review & editing:** Ruiheng Ong, Chirk Jenn Ng, Kalaipriya Gunasekaran, Hang Liu, Wynne Hsu, Mong Li Lee, Ngiap Chuan Tan.

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
