## [Decision Letter · Decision Letter 0]

23 Dec 2024

PONE-D-24-29527Utility of a patient similarity-based digital tool for risk communication to patients with type 2 diabetes mellitus: perspectives from primary care physicians in ambulatory carePLOS ONE

Dear Dr. Ong,

Thank you for submitting your manuscript to PLOS ONE. After careful consideration, we feel that it has merit but does not fully meet PLOS ONE’s publication criteria as it currently stands. Therefore, we invite you to submit a revised version of the manuscript that addresses the points raised during the review process.

We look forward to receiving your revised manuscript.

Kind regards,

Amir Hossein Behnoush

Academic Editor

PLOS ONE

“I have read the journal’s policy and the authors of this manuscript have the following competing interests: WH, MLL and NCT are Principal Investigator and Co-Principal Investigators respectively of the project "JARVIS-DHL: Transforming Chronic Care for Diabetes, Hypertension and HyperLipidemia (DHL) with AI" which is funded by the AI in Health Grand Challenge (Award Number: AISG-GC-2019-001) by AI Singapore (AISG) under the National Research Foundation Singapore.”

3. Please ensure that you include a title page within your main document. You should list all authors and all affiliations as per our author instructions and clearly indicate the corresponding author.

4. We notice that your supplementary figures are uploaded with the file type 'Figure'. Please amend the file type to 'Supporting Information'. Please ensure that each Supporting Information file has a legend listed in the manuscript after the references list.

5. We notice that your supplementary figures and tables are included in the manuscript file. Please remove them and upload them with the file type 'Supporting Information'. Please ensure that each Supporting Information file has a legend listed in the manuscript after the references list.

6. We note that there is identifying data in the Supporting Information file <S3_Table.pdf>. Due to the inclusion of these potentially identifying data, we have removed this file from your file inventory. Prior to sharing human research participant data, authors should consult with an ethics committee to ensure data are shared in accordance with participant consent and all applicable local laws.

-Location data

Please remove or anonymize all personal information, ensure that the data shared are in accordance with participant consent, and re-upload a fully anonymized data set. Please note that spreadsheet columns with personal information must be removed and not hidden as all hidden columns will appear in the published file.

Reviewers' comments:

Reviewer's Responses to Questions

**Comments to the Author**

1. Is the manuscript technically sound, and do the data support the conclusions?

Reviewer #1: Yes

Reviewer #2: Partly

2. Has the statistical analysis been performed appropriately and rigorously? 

Reviewer #1: Yes

Reviewer #2: N/A

3. Have the authors made all data underlying the findings in their manuscript fully available?

Reviewer #1: Yes

Reviewer #2: Yes

4. Is the manuscript presented in an intelligible fashion and written in standard English?

Reviewer #1: No

Reviewer #2: Yes

5. Review Comments to the Author

Reviewer #1: Dear Authors,

This cross-sectional study addresses a vital subject despite having few participants. It considers artificial intelligence tools' great utility in helping physicians better assist their patients. It is a challenge to develop tools that fulfil the multidimensional models of utility. As the authors concluded, using social comparison theory and similarity-based digital tools still requires further work to achieve better results and physician acceptance. Further studies including patients are crucial, especially considering that people who use social comparison may use upward and downward comparisons that may also interfere with the results. However, the whole manuscript deserves a writing review.

Reviewer #2: Introduction:

-consider elaborating on the specific gaps in existing risk communication interventions that this study aims to address.

Method:

- Purposive sampling for diversity is well-documented, but it's unclear if saturation was reached during data collection. Stating explicitly whether saturation influenced the sample size of 11 participants would strengthen the methodological rigor.

-how potential biases from the interviewer (RO) were minimized. Reflexivity could be addressed more explicitly.

- did this study followed any guidelines for qualitative studies?

- how did you choose the PCPs for invitation? how was the sampling?

- were PCPs trained for using PREDICT.AI?

- what was the duration of sessions?

Result:

- it is suggested to mention the specific roles of PCPs in patient care in Table 4

6. PLOS authors have the option to publish the peer review history of their article (what does this mean? ). If published, this will include your full peer review and any attached files.

**Do you want your identity to be public for this peer review?** For information about this choice, including consent withdrawal, please see our Privacy Policy .

Reviewer #1: No

Reviewer #2: No

---

## [Author Response · Author response to Decision Letter 1]

27 Jan 2025

Dear Dr Behnoush and Reviewers,

Utility of a patient similarity-based digital tool for risk communication to patients with type 2 diabetes mellitus: perspectives from primary care physicians in ambulatory care

PONE-D-24-29527

Thank you for giving us the opportunity to improve our manuscript. We appreciate the time and effort you have taken to provide valuable feedback and have revised our manuscript.

We have provided our written responses (responses in blue in the attached file) to each of the points raised in your earlier letter.

EDITOR’S COMMENTS:

Response:

Thank you for highlighting this to us. We have revised the overall manuscript format and file naming.

“I have read the journal’s policy and the authors of this manuscript have the following competing interests: WH, MLL and NCT are Principal Investigator and Co-Principal Investigators respectively of the project "JARVIS-DHL: Transforming Chronic Care for Diabetes, Hypertension and HyperLipidemia (DHL) with AI" which is funded by the AI in Health Grand Challenge (Award Number: AISG-GC-2019-001) by AI Singapore (AISG) under the National Research Foundation Singapore.”

Response:

We confirm that this does not alter our adherence to PLOS ONE policies on sharing data and materials. This has been included into our revised cover letter. We have also updated the number and breakdown of file types into the cover letter.

3. Please ensure that you include a title page within your main document. You should list all authors and all affiliations as per our author instructions and clearly indicate the corresponding author.

Response:

Thank you for highlighting this to us. The title, authors and affiliations have been included in the main document. The previous standalone file containing only the title, authors and affiliations (‘Manuscript Title authors and affiliations.docx’) has been removed.

4. We notice that your supplementary figures are uploaded with the file type 'Figure'. Please amend the file type to 'Supporting Information'. Please ensure that each Supporting Information file has a legend listed in the manuscript after the references list.

Response:

Thank you for highlighting this to us.

The following supplementary files have been renamed and uploaded as a ‘Supporting information’ file:

• ‘S1_Table.pdf’ – renamed ‘S2_Appendix.pdf’

• ‘S2_Table.pdf’ – renamed ‘S3_Appendix.pdf’

• ‘S1_Fig.tif’ – renamed ‘S4_Appendix.tif’

• ‘S2_Fig.tif’ – renamed ‘S5_Appendix.tif’

• ‘S3_Table.pdf’ – renamed ‘S6_Appendix.pdf’

In response to Reviewer #2 question (“Did this study follow any guidelines for qualitative studies?”), we have added the COREQ checklist and uploaded this as ‘S1_Appendix.pdf’ as a ‘Supporting information’ file.

Correspondingly, the legend for the above files have been included in the ‘Supporting information’ section of the main article.

In view of COREQ checklist item 31, we have added ‘Table 1’ – which describes the emergent themes and their respective codes – into the ‘Data analysis’ section of the main article.

We also politely request to retain the following figures and tables in the main article:

• ‘S3_Fig.tif’ (flow diagram of data collection) – renamed ‘Fig1.tif’; image has been removed from main article and uploaded as a ‘Figure’ file after reformatted using the PACE platform

• ‘S4_Table.pdf’ (characteristics of participants) – renamed ‘Table 2’

All of the above changes have been annotated as comments in the revised manuscript copy with tracked changes.

5. We notice that your supplementary figures and tables are included in the manuscript file. Please remove them and upload them with the file type 'Supporting Information'. Please ensure that each Supporting Information file has a legend listed in the manuscript after the references list.

Response:

Thank you for highlighting this to us. We have removed them from the main article. These are detailed in our response to the previous comment.

Correspondingly, the legend for the above files have been included in the ‘Supporting information’ section of the main article.

6. We note that there is identifying data in the Supporting Information file <S3_Table.pdf>. Due to the inclusion of these potentially identifying data, we have removed this file from your file inventory. Prior to sharing human research participant data, authors should consult with an ethics committee to ensure data are shared in accordance with participant consent and all applicable local laws.

-Location data

Please remove or anonymize all personal information, ensure that the data shared are in accordance with participant consent, and re-upload a fully anonymized data set. Please note that spreadsheet columns with personal information must be removed and not hidden as all hidden columns will appear in the published file.

Response:

Thank you for highlighting this to us. We would like to clarify that ‘S3_Table.pdf’ (which has been renamed ‘S6_Appendix.pdf’) contains hypothetical patient case scenarios. No data from actual patients or participants were used. The file also does not contain any form of collected participant data. As such, there is no directly identifiable information. Sex and ethnicity in this context are for the primary care physician participants to enter the hypothetical patient case scenarios into the PERDICT.AI tool.

REVIEWERS’ COMMENTS:

Reviewer #1: Dear Authors,

This cross-sectional study addresses a vital subject despite having few participants. It considers artificial intelligence tools' great utility in helping physicians better assist their patients. It is a challenge to develop tools that fulfil the multidimensional models of utility. As the authors concluded, using social comparison theory and similarity-based digital tools still requires further work to achieve better results and physician acceptance. Further studies including patients are crucial, especially considering that people who use social comparison may use upward and downward comparisons that may also interfere with the results. However, the whole manuscript deserves a writing review.

Response:

Thank you for your esteemed comments. We have revised the writing style, grammar and sentence structure of our manuscript.

Reviewer #2: Introduction:

- consider elaborating on the specific gaps in existing risk communication interventions that this study aims to address.

Response:

Thank you for highlighting this to us. We have included the following in the first paragraph of the ‘Introduction’ to highlight the gaps in risk communication interventions in justifying the conduct of this study:

‘Risk communication interventions for healthcare professionals to convey quantitative risk information of diabetes complications to patients, especially for cardiovascular disease, have been studied with mixed results. For example, French et al. demonstrated that presenting risk information both numerically and graphically improved the accuracy of patients’ risk perceptions but did not lead to behavioural change. Charlson et al. demonstrated no difference in major adverse cardiovascular events when benefits of cardiovascular risk factor modification were framed to patients using present biological age reduction, compared to using future cardiovascular risk reduction. Grover et al. demonstrated that patients were more likely to achieve their lipid targets when their individual cardiovascular risk was presented to them as a ‘cardiovascular age’ (comparing to people of the same age and sex without modifiable risk factors); however, this was only applicable to patients with no pre-existing cardiovascular disease. These highlight a growing need to bridge the knowledge gap on how healthcare professionals can convey risk information in meaningful ways to patients to effectively motivate protective behaviours to reduce the risk of diabetes complications.’

The articles by French et al, Charlson et al, and Grover et al have been cited as reference numbers 10-12 in the revised manuscript.

Method:

- Purposive sampling for diversity is well-documented, but it's unclear if saturation was reached during data collection. Stating explicitly whether saturation influenced the sample size of 11 participants would strengthen the methodological rigor.

Response:

Thank you for highlighting this to us.

The sample size was guided by Hennick and Kaiser who found that saturation is often achieved within 9-17 interviews. Thematic saturation was reached after 9 interviews. No new themes emerged from the last 2 interviews. We have revised the ‘Participants and sampling’ section of the main article as follows:

‘The sample size was guided by Hennick and Kaiser who found that saturation is often achieved within 9-17 interviews. In this study, thematic saturation was reached after 9 interviews and no new themes emerged from the last two interviews.’

The article by Hennick and Kaiser has been cited as reference number 36 in the revised manuscript.

- how potential biases from the interviewer (RO) were minimized. Reflexivity could be addressed more explicitly.

Response:

Thank you for highlighting this to us. We have revised the ‘Study team and reflexivity’ section of the main article to include a write up on ‘reflexivity’ to highlight the potential interviewer’s biases and how they were addressed, as follows:

‘As a practising PCP in a clinic with a high volume of patient visits, RO was cognisant of barriers that PCPs practising in similar environments would face towards adopting new technology. RO reflected on his dual role as a researcher and clinician, and had ongoing discussions on these with CJN and NCT. RO made concerted effort to switch his attention away from his clinician role and immersed himself into the researcher role to maintain distance from his personal views during data collection and analysis. RO remained receptive to understanding positive and negative perspectives given by the PCP participants. PCP participants, especially those of lower seniority to RO, were encouraged during the interviews to share their honest responses and highlight issues faced during usage of PERDICT.AI that would inform the study team on areas to improve the tool. Open-ended questions were used, and leading questions were avoided. Active listening and verbal cues were used to encourage participants to articulate their views. Where needed, paraphrasing and clarification was done to ensure accurate understanding of participants’ responses. CJN was present when RO conducted the first interview, and a debriefing was done post-session.’

- did this study followed any guidelines for qualitative studies?

Response:

The study followed the consolidated criteria for reporting qualitative studies (COREQ). We have included this in the ‘Study design and setting’ section of the main article. We have also added the COREQ checklist and uploaded this as ‘S1_Appendix.pdf’ as a ‘Supporting information’ file.

Additional details in the manuscript from the COREQ checklist items have been annotated as comments in the revised manuscript copy with tracked changes.

The article describing COREQ (by Tong et al.) has been cited as reference number 35 in the revised manuscript.

- how did you choose the PCPs for invitation? how was the sampling?

Response:

PCPs were recruited by purposive sampling with maximum variation to capture a diverse range of seniorities, with participants representing early-career (Medical Officers), mid-career (Family Physicians) and late-career (Consultants) stages.

We have revised the ‘Participants and sampling’ section of the main article as follows:

‘PCPs were recruited by purposive sampling with maximum variation to capture a diverse range of seniorities, with participants representing early-career (Medical Officers), mid-career (Family Physicians) and late-career (Consultants) stages.’

- were PCPs trained for using PREDICT.AI?

Response:

A user guide on PERDICT.AI was provided to each PCP participant during invitation to participate in the study. Training on using PERDICT.AI was not conducted as the intent was for participants to use PERDICT.AI for the first time as a naïve user.

We have revised the ‘Participants and sampling’ section of the main article as follows:

‘Training on using PERDICT.AI was not conducted, as the intent was for participants to use PERDICT.AI for the first time as a naïve user.’

- what was the duration of sessions?

Response:

The duration of each session was between 17-48 minutes. We have revised the ‘Data collection’ section of the main article as follows:

‘Each participant took between 17 and 48 minutes to complete the case scenarios. ’

Result:

- it is suggested to mention the specific roles of PCPs in patient care in Table 4

Response:

Thank you for the suggestion. We would like to clarify that the PERDICT.AI case scenarios are hypothetical cases and no actual patient care was delivered by the PCP participants as part of the study.

Nevertheless, we feel it would be helpful to the reader to know that during routine clinic days (when not participating in the study), PCPs across the three clinics provide ambulatory primary care to patients with T2DM. This includes the diagnosis of T2DM, monitoring of glycaemic control, and the initiation and adjustment of anti-diabetic agents and insulin. We have included this into the ‘Study design and setting’ section of the main article as follows:

‘This includes the diagnosis of T2DM, monitoring of glycaemic control, and the initiation and adjustment of anti-diabetic agents and insulin.’

‘S4_Table.pdf’ (Table 4) has been renamed ‘Table 2’.

In addition, we have organised the ‘Results’ section into two subsections ‘Participants characteristics’ and ‘Principal findings’ (describing the three themes). We have revised the description

---

## [Decision Letter · Decision Letter 1]

12 Feb 2025

Utility of a patient similarity-based digital tool for risk communication to patients with type 2 diabetes mellitus: perspectives from primary care physicians in ambulatory care

PONE-D-24-29527R1

Dear Dr. Ong,

We’re pleased to inform you that your manuscript has been judged scientifically suitable for publication and will be formally accepted for publication once it meets all outstanding technical requirements.

Kind regards,

Amir Hossein Behnoush

Academic Editor

PLOS ONE

Additional Editor Comments (optional):

Reviewers' comments:

Reviewer's Responses to Questions

**Comments to the Author**

1. If the authors have adequately addressed your comments raised in a previous round of review and you feel that this manuscript is now acceptable for publication, you may indicate that here to bypass the “Comments to the Author” section, enter your conflict of interest statement in the “Confidential to Editor” section, and submit your "Accept" recommendation.

Reviewer #1: All comments have been addressed

Reviewer #2: (No Response)

2. Is the manuscript technically sound, and do the data support the conclusions?

Reviewer #1: Partly

Reviewer #2: (No Response)

3. Has the statistical analysis been performed appropriately and rigorously? 

Reviewer #1: N/A

Reviewer #2: (No Response)

4. Have the authors made all data underlying the findings in their manuscript fully available?

Reviewer #1: Yes

Reviewer #2: (No Response)

5. Is the manuscript presented in an intelligible fashion and written in standard English?

Reviewer #1: Yes

Reviewer #2: (No Response)

6. Review Comments to the Author

Reviewer #1: Dear Authors,

This cross-sectional study addresses a vital subject despite few participants, reinforcing the need for further studies with a more significant sample and addressing some questions of clinical utility. In response to reviewers' comments, some modifications have improved the manuscript, but it is important to reconsider the saturation. The sample size reported by Hennick and Kaiser in their review was achieved considering homogeneous samples. This study included participants of 28 to 57 years of age and clinical experience of less than one year to more than eleven years, which means different generations and expertise that may have very different views, experiences and knowledge with artificial intelligence tools and clinical practice. It is also important to comment on the limitations of purposive sampling, which may present inherent bias. It considers artificial intelligence tools' great utility in helping physicians better assist their patients. It is a challenge to develop tools that fulfil the multidimensional models of utility. Using social comparison theory and similarity-based digital tools still requires further work to achieve better results and physician acceptance.

Reviewer #2: (No Response)

7. PLOS authors have the option to publish the peer review history of their article (what does this mean? ). If published, this will include your full peer review and any attached files.

**Do you want your identity to be public for this peer review?** For information about this choice, including consent withdrawal, please see our Privacy Policy .

Reviewer #1: No

Reviewer #2: No

---

## [Editor Report · Acceptance letter]

PONE-D-24-29527R1

PLOS ONE

Dear Dr. Ong,

I'm pleased to inform you that your manuscript has been deemed suitable for publication in PLOS ONE. Congratulations! Your manuscript is now being handed over to our production team.

Kind regards,

on behalf of

Dr. Amir Hossein Behnoush

Academic Editor

PLOS ONE